

# The Earth Topography 2022 (ETOPO 2022) Global DEM dataset

Michael MacFerrin[1,2], Christopher Amante[1,2], Kelly Carignan[1,2], Matthew Love[1,2], Elliot Lim[1,2]

[1]Cooperative Institute for Research in Environmental Sciences, University of Colorado, Boulder, 80309, United States of America
[2]National Centers for Environmental Information, National Oceanic and Atmospheric Administration, Boulder, Colorado, 80309, United States of America

*Correspondence to*: Michael J. MacFerrin (michael.macferrin@colorado.edu)

**Abstract.** Here we present Earth TOPOgraphy (ETOPO) 2022, the latest iteration of NOAA's global, seamless topographic-bathymetric dataset. ETOPO1, NOAA's prior release at 1-arc-minute resolution, has been a widely-used benchmark global digital elevation model (DEM) since its initial release in 2009 (Amante and Eakins, 2009). Tsunami forecasting, modeling, and warning systems critically rely upon accurate topographic and bathymetric data to predict and reproduce water movement across global ocean surfaces, wave heights at the coastline, and subsequent land inundation. ETOPO 2022 is an updated topographic-bathymetric dataset at 15-arc-second global resolution that incorporates bare-earth datasets with forests and buildings removed. ETOPO 2022 integrates more than a dozen source datasets for land topography, sea bathymetry, lake bathymetry, and ice-sheet bed elevation data, all of which have been carefully evaluated for quality, accuracy, and seamless integration. We evaluate the relative and absolute vertical accuracies of all land-elevation input datasets, as well as the final ETOPO 2022 tiles, using a geographically optimized, independent database of bare-earth elevation photons from NASA's ICESat-2 satellite mission over the calendar year 2021. Measured against more than 960 billion lidar measurements from ICESat-2 that span nearly the entire globe, ETOPO 2022 measures a global RMSE of 7.17 m. ETOPO 2022 is publicly available in both ice surface and bedrock versions that portray either the top layer of the ice sheets covering Greenland and Antarctica, or the bedrock below, and both versions are also available in GeoTiff and NetCDF formats in 15x15° tiles, as well as global tiles at 30- and 60-arc-second resolutions. ETOPO 2022 provides a new, publicly available, seamless, globally validated elevation dataset to meet the present and future needs of the scientific global hazard and mapping communities.

## 1 Introduction

Earth scientists and modelers often rely upon accurate, large-scale models of Earth's surface elevation for a variety of earth-modeling applications. The National Centers for Environmental Information (NCEI) at the National Oceanic and Atmospheric Administration (NOAA) has long produced seamless earth topographic datasets by combining topographic and bathymetric data from a variety of sources. The "Earth TOPOgraphy" (ETOPO) datasets have been produced at 5-arc-minute, 2-minute, and 1-minute horizontal resolutions covering the entire earth surface. ETOPO 2022 provides an updated global elevation at a





refined spatial resolution of 15-arc-second from the ETOPO1 (1-arc-minute) dataset last released in 2009. Primary end-users
of ETOPO are coastal hazard and tsunami modelers; however, ETOPO is used as a baseline dataset in thousands of scientific
papers, data products, and references worldwide (e.g. Friedlingstein et al., 2020; Schmidtko et al., 2017; Woodruff et al., 2013).
**2 Data Description**
**2.1 General Description and File Formats**
ETOPO 2022 is a full coverage, seamless, gridded topographic and bathymetric elevation dataset. ETOPO 2022 is an updated,
higher-resolution version of previously released ETOPO5 (5 arc-minute), ETOPO2 (2 arc-minute), and ETOPO1 (1 arc-
minute) global grids. For further use in this document, references to "ETOPO" refer to the ETOPO 2022 release. References
to any previous ETOPO grids (ETOPO1, ETOPO5, etc) use the specific version names.
ETOPO is released as a global-coverage dataset comprised of 288 individual 15x15 degree tiles (latitude/longitude) at 15-arc-
second geographic resolution. The tiles are provided in GeoTiff and Network Common Data Form (NetCDF) formats, with
identical information provided in each format. An additional 62 tiles have "bed" versions that provide bedrock elevations under
the surface of the Greenland and Antarctic ice sheets. All tiles are in horizontal WGS84 geographic coordinates (EPSG:4326)
and vertically referenced in meters relative to the Earth Gravitational Model of 2008 (EGM2008) geoid surface (EPSG:3855).
Each tile comes with an accompanying integer Source ID ("sid") tile specifying from which source dataset each ETOPO
elevation was derived (see Section 3 Input Datasets and Pre-processing), as well as an accompanying "geoid" tile for converting
EGM2008 geoid heights into WGS84 ellipsoid elevation heights (EPSG:4979). Since most other geoid, ellipsoid, and/or tidal
vertical datums are defined by grids in reference to the WGS84 ellipsoid, this eases the conversion of ETOPO 2022 tiles into
other vertical reference datums of the user's choice. For most purposes, EGM2008 is an adequate approximation of mean sea
level at the 15 arc-second resolution of ETOPO.
**2.2 File Naming Convention**
ETOPO 2022 tiles are named in the following manner:

54                    **ETOPO_2022_v[#]_[RR]s_[N][YY][W][XXX][_suffix][.tif]**


with the following information in place of the brackets []:
[#] - Version number of the release. In this case, version 1.
[RR] - Data tile resolution (15, 30, 60), in arc-seconds
[N] - "N" or "S", for Northern or Southern hemisphere
[YY] - 2-digit latitude of tile's northern (top) border, absolute value



[W] - "W" or "E", for Eastern or Western hemisphere
[XXX] - 3-digit longitude of the tile's western (left) border, absolute value
[_suffix] - "_surface": surface elevations; "_bed": bed elevations, "_sid": source id numbers, "_geoid": geoid heights.
[.tif] - File extension: ".tif" (GeoTiff) or ".nc" (NetCDF) formats.

For example, a tile named

**ETOPO_2022_v1_15s_N60W045_bed.tif**

is a GeoTiff file with a resolution of 15 arc seconds, and its upper-left corner is located at a latitude of 60 degrees North and a
longitude of 45 degrees West. In this case, the file contains data on bedrock elevations beneath the surface of either the
Greenland or Antarctic ice sheets.
**2.3 Geoid Conversion**
To convert a given tile from EGM2008 to WGS84-referenced elevations (which can be easily converted to other vertical
datums), add the values of the elevation tile to the geoid-height tile:

ETOPO Elevation (EGM2008) + GEOID = WGS84 Elevation                                              (1)

To enable easy conversion between vertical elevation reference grids, geoid files are distributed alongside each ETOPO
elevation tile. In ice surface and bedrock versions, single global tiles are also provided at 30- and 60-arc-second (i.e., 1-arc-
minute) resolutions in both GeoTiff and NetCDF format. 30- and 60-second grids were downsampled from the 15-arc-second
elevation tiles for more general uses, and do not have accompanying SID tiles.
**3 Input Datasets and Pre-processing**
Table 1 lists the datasets that contributed elevation data in the ETOPO product. Other data sources were assessed and evaluated,
but were not included in the final ETOPO 2022 data product. The source name acronyms for each dataset are defined in the
sections following Table 1.

**Table 1.** Metadata of the ETOPO source datasets.

| Source Name | Vertical Datum (as distributed) | Layer source ID | Creator | Primary Use | % total coverage, surface | % total coverage, bed |
|---|---|---|---|---|---|---|





| GEBCO 2022 | MSL | 1 | GEBCO Compilation Group (2022) | Sea bathymetry, base layer, large lake bathymetry | 58.78 % | 49.66 |
|---|---|---|---|---|---|---|
| GEBCO 2022 Sub-ice | MSL | 2 | GEBCO Compilation Group (2022) | Sea bathymetry (sub-ice, polar regions) | 0.00 % | 8.40 % |
| NOAA Estuarine DEMs | various | 3 | NOAA/NCEI (archived) | Sea bathymetry | <0.01 % | <0.01 % |
| NOAA Regional DEMs | various | 4 | NOAA/NCEI (archived) | Sea bathymetry | 0.22 % | 0.22 % |
| GMRT 4.0 | MSL | 5 | GMRT.org, Lamont-Doherty Earth Observatory | Sea bathymetry | 6.75 % | 6.73 % |
| Shallow Bathymetry Everywhere | EGM2008 geoid | 6 | Oregon State University | Sea bathymetry | <0.01 % | <0.01 % |
| BlueTopo | NAVD88 | 7 | NOAA OCS | Sea bathymetry | 0.05 % | 0.05 % |
| BOEM Gulf of Mexico Bathymetry | MSL | 8 | BOEM | Sea bathymetry | 0.03 % | 0.03 % |
| Copernicus DEM 30m | EGM2008 geoid | 9 | European Space Agency | Land topography | 10.60 % | 0.12 % |
| FABDEM | EGM2008 geoid | 10 | European Space Agency and Bristol University | Land topography | 23.28 % | 22.46 % |
| GEBCO Lake Depths | MSL | 11 | GEBCO Hydrolakes outlines and GEBCO elevations | Global surveyed lake depths (for very large lakes) | 0.12 % | 0.12 % |
| BedMachine | EIGEN-6C4 geoid | 12 | NASA | Ice sheet bed topography | 0.00 % | 12.05 % |





| CUDEM | various | 13 | NOAA Coastal DEM Team | Land Topography and sea bathymetry (US & Territories) | 0.16 % | 0.16 % |
|---|---|---|---|---|---|---|


Figures 1 and 2 show the distribution of source datasets across the ETOPO 2022 product for the surface products (Figure 1)
and bed products (Figure 2).

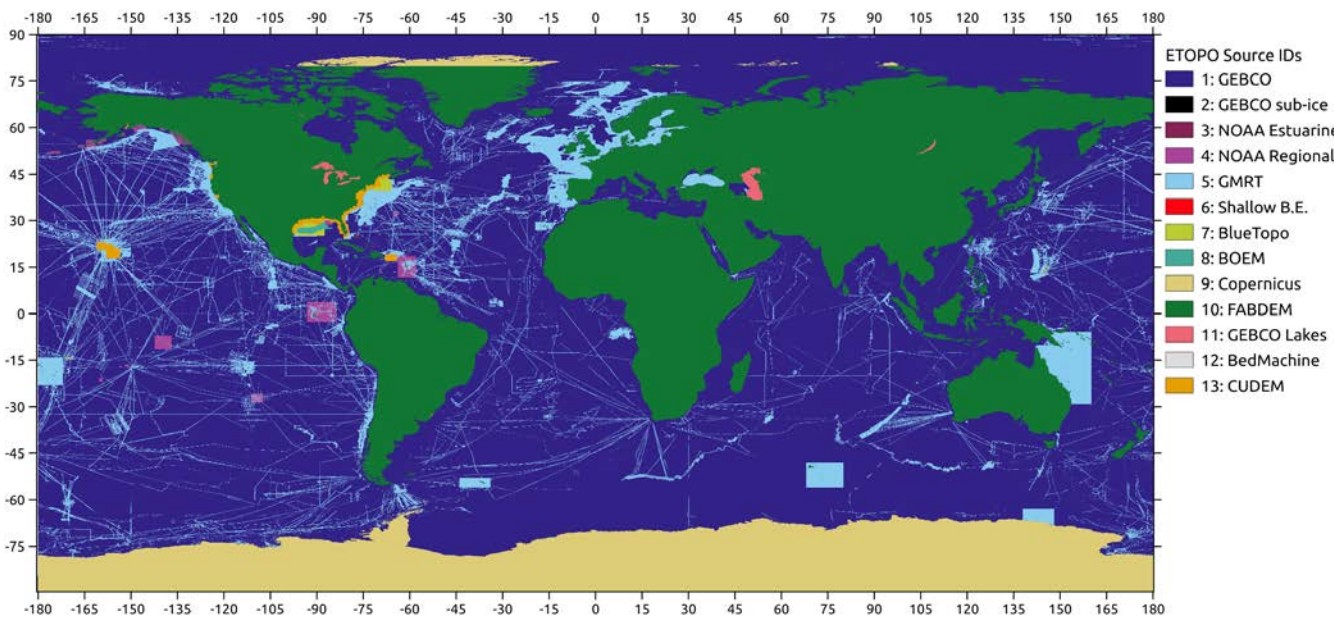


**Figure 1:** Map of ETOPO 2022 Surface source datasets.



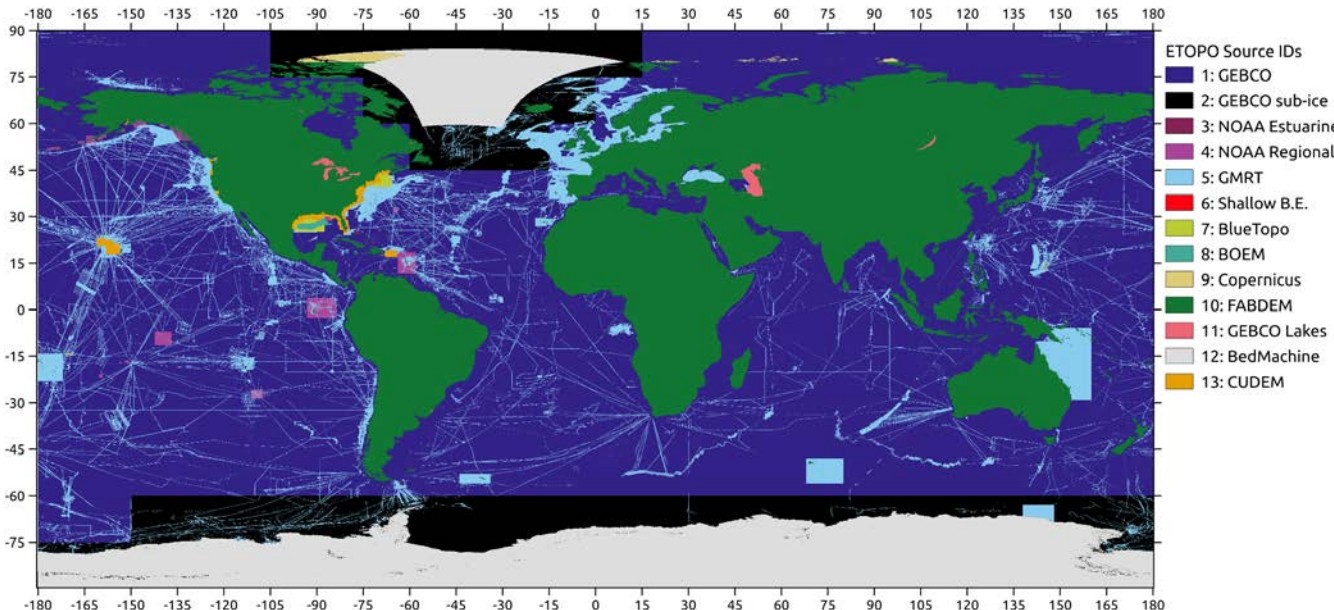

**Figure 2:** Map of ETOPO 2022 Bedrock source datasets.

The following datasets (Table 2) were not directly included in the ETOPO tiles, but were used for the development, production, and/or validation of the source data layers, as described in further sections.

**Table 2:** Datasets used in ETOPO production and validation but not contributing directly to ETOPO elevation values

| Source Name | Vertical Datum | Creator | Primary Use |
|---|---|---|---|
| ICESat-2 - ATL03 and ATL08 | EGM2008 / WGS84 | NASA | Photon elevation data for DEM evaluation |
| Hydrolakes | n/a | HydroSHEDS | Global vector outlines of inland water bodies |
| National Hydrography Dataset (NHD) | n/a | U.S. Geological Survey | Vector outlines of North American inland water bodies |
| World Settlement Footprint 2015 | n/a | (Marconcini, et al., 2020) | Heavy-urban-area footprints (masked during ICESat-2 validation) |

We performed the following pre-processing steps on each dataset before incorporating into the ETOPO 2022 product.



## 3.1 GEBCO 2022

The General Bathymetric Chart of the Oceans (GEBCO) is an annually-produced global elevation product derived from a global consortium of institutions collaborating on the SEABED 2030 project, with the primary aim of mapping the world's ocean bathymetry in its entirety by the year 2030 (Mayer et al., 2018). GEBCO global elevation grids are produced at 15-arc second resolution and incorporate a mix of data sources, including sonar soundings, lead-line measurements, and interpolated gravimetry data for bathymetry. ETOPO uses the global GEBCO grids as a "base layer", using GEBCO data where other direct measurements are not available. The land-based portions of the GEBCO global grids are based upon reprocessed NASA Shuttle Radar Topography Mission (SRTM) data collected in February 2000 (Rodríguez et al., 2006). Although ETOPO 2022 includes GEBCO in its base-layer even over land, the land-based portions of the ETOPO grid are based primarily on modern satellite radar-derived measurements, and as such, GEBCO data is not used over land for a majority of the ETOPO product.

For a small set of large inland water bodies, GEBCO contains surveyed bathymetry data derived from other sources. For each of the following lakes, raster masks for the lake areas were produced from digitizing outlines from the vector HydroLakes dataset (Messager et al., 2016), part of the HydroSHEDs database of global land hydrography data. A separate data layer incorporating just the lake bathymetry from GEBCO was produced and given a higher topographic source ID number than the primary land-based topographic datasets such as CopernicusDEM and FABDEM, so that lake bathymetries supersede other surface topography datasets. The large lakes and coastal estuarine areas in which GEBCO includes plausible lake bathymetry are outlined in Table 3. These lakes were not chosen because they were inherently the biggest in the world (although several of them are the largest lakes on Earth by area), but rather because it was determined that GEBCO contained plausible bathymetry for these lakes, while using a "flat surface" for remaining lakes worldwide. Bathymetries of other large lakes may be included in further updates to the ETOPO data product.

**Table 3:** Large lakes and estuarine areas from which approximate bathymetry was pulled from GEBCO.

| Name | Center Location (Lat, Lon) | Approximate Area (km$^2$) | ETOPO Tile ID(s) |
|---|---|---:|---|
| Caspian Sea | 41.9 °N, 50.6 °E | 371,000 | N45E045, N30E045 |
| Superior | 47.8 °N, 88.1 °W | 82,103 | N45W105, N45W090 |
| Huron | 44.8 °N, 82.4 °W | 59,600 | N45W090, N30W090 |
| Michigan | 44.1 °N, 87.0 °W | 58,030 | N45W090, N30W090 |
| Baikal | 53.3 °N, 108.0 °E | 31,722 | N45E105, N45E090 |



| Erie | 42.2 °N, 81.3 °W | 25,740 | N30W090 |
| Ontario | 43.6 °N, 78.0 °W | 18,960 | N30W090 |
| Laguna Merin | 32.8 °S, 53.2 °W | 4,500 | S45W060 |
| Melville | 53.8 °N, 59.4 °W | 3,069 | N45W075, N45W060 |
| Baker | 64.2 °N, 95.4 °W | 1,887 | N60W105 |
| Bras d'Or | 45.9 °N, 60.8 °W | 1,100 | N45W075 |
| Selawik | 66.5 °N, 160.7 °W | 1,050 | N60W165 |

127

**3.2 NOAA Estuarine DEMs**

In 2018, NOAA updated the National Ocean Service's Estuarine Bathymetry DEMs, gridded representations of bathymetry for various estuaries in the United States, which were initially created in 1998 by the now defunct NOS Special Projects Office. The Estuarine DEMs (National Centers for Environmental Information (NCEI), 2020) provide nearshore and up-river bathymetry for multiple US-based estuarine areas, provided in Mean Low-Low Water (MLLW) tidal elevations. Although these data still represent the "best available" gridded depictions of bathymetry in some locations, they are primarily based on antiquated historical data and do not include many modern survey data, in particular, high-resolution Bathymetric Attributed Grid (BAG) format hydrographic data. The only available data digitized before 1997 were used in the original project. The majority of Estuarine DEMs were included in ETOPO, while several others were omitted where higher-quality data was available from other sources. Most NOAA Estuarine datasets were superseded by other more recent datasets and thus incorporate a small area of the final ETOPO product (less than 0.001 % of global land area).

**3.3 NOAA Regional DEMs**

Before the initiation of NOAA's Continuously-Updated Digital Elevation Model (CUDEM) program in 2014 (Amante et al., 2023), the NOAA Coastal Digital Elevation Model team produced numerous regional, integrated topographic-bathymetric DEMs covering various regions within the coastal waters of the United States. These Regional DEMs (NCEI, 2020) are derived from a variety of available data sources at the time of creation and are output in various tidal vertical datums to fit the needs of individual organizations and groups (both internal and external to NOAA) that requested coastal DEMs. The regional DEMs are available on NOAA's THREDDS Catalog at https://www.ngdc.noaa.gov/thredds/catalog/regional/catalog.html. Similar to the NOAA Estuarine DEMs, some individual files were omitted from ETOPO due to the availability of higher-quality data in



a specific region. In some areas, specific sub-areas were filtered out from individual regional DEMs due to artifacts, prior to
inclusion in ETOPO 2022. NOAA NCEI-created topographic and bathymetric data newer than the Regional DEMs are
included in the high-resolution CUDEM layer (Section 3.11).
**3.4 GMRT v4.0**
The Global Multi-Resolution Topography Synthesis project (Ryan et al., 2009) maintains a database of gridded high-resolution
topographic and bathymetric datasets around the world. They are produced and distributed at multiple gridded resolutions.
GMRT primarily focuses on the ingestion and processing of ship-based multibeam sonar data acquired by the United States
Academic Research Fleet (ARF). Additionally, GMRT utilizes multibeam sonar and other relevant sources and projects where
available. Elevations over land are derived from the United States National Elevation Dataset (NED) and NASA Advanced
Spaceborne Thermal Emission and Reflection Radiometer (ASTER) global DEM. Other datasets were used for land elevations
in ETOPO 2022, and GMRT is primarily used where multi-beam sonar data exists. ETOPO 2022 made use of GMRT 4.0 data
as it existed in June 2022.

Some regions in the GMRT bathymetry data—specifically regions that were not derived from multibeam sonar—contained
artifacts that did not reflect the true bathymetry in those locations. When such artifacts were found, we manually generated
bounding boxes around such regions and filtered them out from the GMRT data (filling with no-data values) before ingesting
GMRT into the ETOPO project. These "omitted" regions from GMRT data are outlined in the data file
"GMRT_omitted_regions_15s.csv" included in this dataset.
**3.5 Shallow Bathymetry Everywhere**
The Shallow Bathymetry Everywhere project (Forfinski-Sarkozi and Parrish, 2019) maps shallow-water bathymetry using
optical image techniques, primarily using the Landsat-8 satellite with machine learning techniques and validated against
existing bathymetry surveys and remotely-sensed ICESat-2 lidar data (Forfinski-Sarkozi, et al, 2019). At publication time, the
dataset encompasses 12 specific regions worldwide. Eleven regions covering shallow ocean bathymetry were included in
ETOPO 2022 while excluding one dataset over an inland lake.
**3.6 BlueTopo**
BlueTopo is a suite of gridded coastal bathymetry datasets at nested resolutions released by the NOAA Office of Coast Survey
(OCS) and distributed publicly (U.S. Office of Coast Survey, 2022). BlueTopo surveys were used where the data was extracted
from measurements, whereas regions of interpolated data (usually drawn as triangular irregular networks between isolated
survey points) were omitted from ETOPO. Additionally, some data was omitted that was sourced from older datasets (older
regional DEMs, e.g.) for which more recent data was available from other sources. The BlueTopo tiles come in nested
resolutions from 16 m to 2 m grid-cell spacings, in powers of 2. Higher-resolution tiles were weighted above lower-resolution



tiles where both existed, favoring the higher-resolution data when subsetting data into ETOPO grid cells. BlueTopo tiles were
re-gridded from Universal Transverse Mercator (UTM) projections into the World Geodetic Survey 1984 geographic grids,
and vertically transformed from the North American Vertical Datum 1988 (Navd88) into EGM 2008 elevations before
inclusion in ETOPO.

**3.7 Bureau of Ocean Energy Management (BOEM) Gulf of Mexico Bathymetry**

BOEM released a high-resolution bathymetric map of the northern Gulf of Mexico region from active seismic acoustic surveys
in 2017 (Kramer and Shedd, 2017). The BOEM gridded dataset consists of 1.4 billion grid cells at 40 by 40 foot horizontal
resolution, with depths relative to mean sea level. BOEM is publicly available for download.  The two BOEM data grids
(covering the Eastern and Western Gulf of Mexico) were each projected horizontally into WGS84 geographic coordinates
before inclusion in ETOPO.

**3.8 Copernicus DEM 30 m**

The Copernicus DEM 30 m global digital elevation model (GLO-30)(The European Space Agency, 2022) was produced by
the European Space Agency's Copernicus program from spaceborne altimetric radar measurements. GLO-30 is provided
worldwide with the exception of 25 1-degree tiles in the Armenia and Azerbaijan regions. A recent study compared the
accuracies of multiple global land-elevation models (The European Space Agency, 2022), and found that Copernicus provided
the lowest vertical errors compared against high-accuracy airborne lidar datasets in select study areas. The GLO-30 product is
a "digital surface model" indicating it measures the top of tree canopies and buildings rather than bare-Earth elevations, which
may result in biases when compared to bare-earth elevation datasets. Copernicus was used as the primary land-elevation layer
in the polar regions (Arctic and Antarctic) where forests and urban areas are rare or nonexistent.

**3.9 FABDEM v1.0**

The Forest and Buildings Removed Copernicus DEM (FABDEM) (Hawker et al., 2022) combines the Copernicus DEM GLO-
30 product with canopy data products and modeling to produce a simulated global bare-earth Digital Terrain Model (DTM).
Satellite-derived forest canopy height measurements come from NASA's Global Ecosystem Dynamics Investigation (GEDI)
mission (Dubayah et al., 2020) Global Forest Canopy Height 2019 product (Potapov et al., 2021) as well as canopy elevations
derived from ICESat-2 lidar measurements (Neuenschwander and Magruder, 2019), built-environment footprints from the
World Settlement Footprint (WSF) (Marconcini et al., 2020) and numerous others data layers to produce a model for canopy
and building elevation biases within the Copernicus 30 m GLO-30 product. Correcting for these biases, they produced the
FABDEM v1.0 product, which was shown to reduce the errors in their respective study areas against reference DEMs produced
by high-accuracy airborne lidar. FABDEM is available for land elevations between 60 °S and 80 °N latitudes and is used in
ETOPO where available. Copernicus DEM was used in the polar regions south of 60 °S latitude and north of 80 °N. Since the



release of ETOPO 2022, FABDEM has been updated version 1.2 to further reduce biases and errors, especially in steeply
sloped regions (Neal et al., 2023).

**3.10 BedMachine Greenland and Antarctica**

The BedMachine Greenland version 5 (Morlighem et al., 2017) and BedMachine Antarctica version 3 (Morlighem, 2020)
datasets were used to produce the "bedrock" versions of ETOPO with the Greenland and Antarctic ice sheets removed.
BedMachine derives gridded ice thickness data from a combination of NASA airborne radar-sounding measurements and a
novel interpolation method that combines ice-flow velocities and model calculations to conserve mass across flowlines of
glaciers to provide likely estimates of interpolated bed elevations between direct radar measurements. BedMachine elevations
were converted from the Eigen-6C4 geoid to the EGM 2008 vertical references, and converted from polar stereo projections
into WGS84 geographic grids for inclusion in ETOPO. It was found that in offshore waters surrounding Greenland,
BedMachine derives much of its bathymetric elevation data from the same sources as GEBCO, and thus was used without
masking for bed elevations of the Greenland ice sheet and surrounding ocean waters together. Although BedMachine
Antarctica and BedMachine Greenland are different datasets, they do not overlap spatially, and were combined into the same
dataset layer for ETOPO (Table 1). BedMachine data is only used in the ETOPO 2022 "bedrock" elevation products
overlapping the Greenland and Antarctic ice sheets, and are unused in the ETOPO "surface" tiles.

**3.11 CUDEM**

The Continuously Updated Digital Elevation Model (CUDEM) framework at NOAA produces high-resolution coastal
topographic and bathymetric bare-earth DEMs for U.S. states and territories (Amante et al., 2023). CUDEM combines a suite
of airborne, spaceborne, and shipborne data to produce seamless topographic and bathymetric datasets in coastal areas for
coastal hazard modeling and management, in a framework that allows frequent on-demand updates after significant coastal
changes. The CUDEMs are currently the highest-resolution, seamless depiction of the entire U.S. Atlantic and Gulf Coasts in
the public domain; coastal topographic-bathymetric DEMs have a spatial resolution of 1/9th arc-second (~3 m) and offshore
bathymetric DEMs coarsen to 1/3rd arc-second (~10 m; Amante el al., 2023). CUDEMs also provide high-resolution DEM
coverage for Hawaii, American Territories, and portions of the U.S. Pacific Coast. CUDEM tiles generated prior to August
2022 were included in ETOPO 2022.

**4 Methods**

**4.1 CUDEM Stacks**

The Continuously Updated Digital Elevation (CUDEM) framework (Amante et al., 2023) at the NOAA Centers for
Environmental Information (NCEI) and the Cooperative Institute for Research in Environmental Sciences (CIRES) at the
University of Colorado, build and provide a series of Python based software tools for the efficient building of seamless DEM



data products from a variety of sources. ETOPO was built primarily using the CUDEM "stacks" module, which stacks raster
layers such as those listed in Table 1 from a variety of datasets (in various horizontal projections) using weights provided by
the user. The stacks module computes output DEMs using a weighted average of the source datasets overlapping a given output
grid-cell, or if the "supersede" flag is set, uses the highest-ranked dataset of all data overlapping a given grid-cell. ETOPO was
built from the source datasets listed in Table 1 using the stacks module with the supercede flag set. Source data that was at
equal or lower-resolution than the output ETOPO grid cells were interpolated using bilinear interpolation from the source
dataset. Source data that was higher-resolution than the ETOPO grid cells were interpolated using an average of overlapping
grid cells.

**4.2 Vertical Datum Transformations**

Gridded input datasets whose vertical reference datum differed from the EGM2008 geoid, and for which transformation grids
are available, were transformed vertically into EGM2008 reference elevations using the NOAA VDatum Tool, version 4.4
(US Department of Commerce, 2022). BedMachine data products (Greenland and Antarctica) were vertically transformed
from the EIGEN-6C4 geoid into WGS84 ellipsoid elevations using the geoid grids included with BedMachine, and then into
EGM2008 using VDatum. In some individual cases (such as NOAA Estuarine and Regional DEMs), DEMs in local tidal
datums (such as "mean-low-low-water" [MLLW]) were converted using interpolated grids from local tide stations, and from
there to EGM2008. Some datasets presented as being referenced to "MSL" were not referenced to any global datum, and these
were unable to be mathematically converted to EGM2008. These datasets were primarily used in off-shore regions where the
differences between MSL and the EGM2008 geoid heights are far less than the uncertainties in the bathymetry measurements
themselves. In such cases, MSL-referenced data was included unchanged in ETOPO 2022. Any uncertainties added from this
implicit non-conversion of data are included in the uncertainty estimates of the ETOPO product.

**4.3 Coastline Masking of Copernicus and FABDEM**

Copernicus and FABDEM provided the majority of land-elevation data for the ETOPO 2022 product. Both datasets contain
zero values over ocean waters, which are treated as "NoData." When Copernicus and FABDEM are resampled from their
native 1-arc-second resolutions to the ETOPO 2022 15-arc-second resolutions, it can cause the shoreline to "creep" by 1 pixel,
because any 15-arc-second grid-cell would be classified as coming from Copernicus or FABDEM if even a fraction of a single
1-second grid cell from those datasets were included anywhere in the ETOPO grid-cell. To avoid this, both Copernicus and
FABDEM were resampled into the ETOPO 15-arc-second grid using both "mean" and "nearest-neighbor" interpolation
methods. The nearest-neighbor produced dataset only contained data if the source dataset overlapped with the center of the
ETOPO-grid cell, providing a more realistic shoreline outline than using the "mean"-derived data. The mean-derived data was
produced for the elevations is provided, but the coastline of mean values was masked using the "nearest neighbor" derived
data, so that a mean elevation was produced only if Copernicus or FABDEM overlapped with the center of the ETOPO grid
cell. These resampled and masked tiles were used in final production of the ETOPO tiles.






## 4.4 Production of 30- and 60-second tiles

The ETOPO 15-arc second dataset is available in 288 global tiles at 15° latitude and longitude intervals. For users with global applications who do not need the highest resolution, ETOPO is produced in 30- and 60-second (1-arc-minute) resolutions in single global files, in both surface and bedrock versions. The 30- and 60-second global tiles were produced by mean-interpolating and stitching the 15-second ETOPO tiles into a single file. Since the lower-resolution files were generated by averaging the higher-resolution ETOPO, no source ID (sid) files are produced for the ETOPO 30- and 60-second versions

## 5 Validation Methods

The Ice, Cloud, and land Elevation Satellite 2 (ICESat-2) is a photon-counting spaceborne altimetric lidar. ICESat-2 data was used to rank datasets as well as validate the ETOPO 2022 product over land. ICESat-2 photons from the calendar year 2021 were assimilated and used to assess the bare-earth elevations of land photons over grid cells that underlie ICESat-2 orbit passes. A small number of ICESat-2 granules were discarded due to the presence of data artifacts.

Figure 3 shows a point cloud of a single ICESat-2 orbit track over the northeast U.S. from June 1, 2022. By linking ICESat-2's ATL03 v5 Photon data product (Neumann, 2021) with its ATL08 Land and Vegetation Elevation (Neuenschwander and Pitts, 2019) data product, we classified photons as land-surface, canopy, canopy top, and noise. Atmospheric/noise photons, seen as "grey" in Figure 3, were discarded. Although canopy and canopy-top photons were used for assessing approximate vegetation cover, they were not used directly in validation processing against the ETOPO bare-earth dataset. Only photons that were classified as land or ice-surface in the ATL03 product, with a "high" confidence level, were included. Since ETOPO is a bare-earth elevation product and ICESat-2 does not filter out photons reflected from the tops of urban structures, validating ETOPO in regions with high rooftops introduces a false negative bias in ETOPO validations using ICESat-2. We used the World Settlement Footprint (WSL) dataset to filter out regions of heavy-urban building cover to help alleviate this bias. In higher-resolution validations, we use the OpenStreetMap database to filter out photons at individual building levels, but such a mask was infeasible at ETOPO's 15 arc-second resolution. Lastly, we filtered out photons that likely reflected off regions of open water using the US National Hydrography Dataset Plus (NHDplus) (Moore et al., 2019) as well as the global HydroLakes (Khazaei et al., 2022) dataset. Best attempts were made to only validate ETOPO against ICESat-2 over grid-cells that represent the land topography.

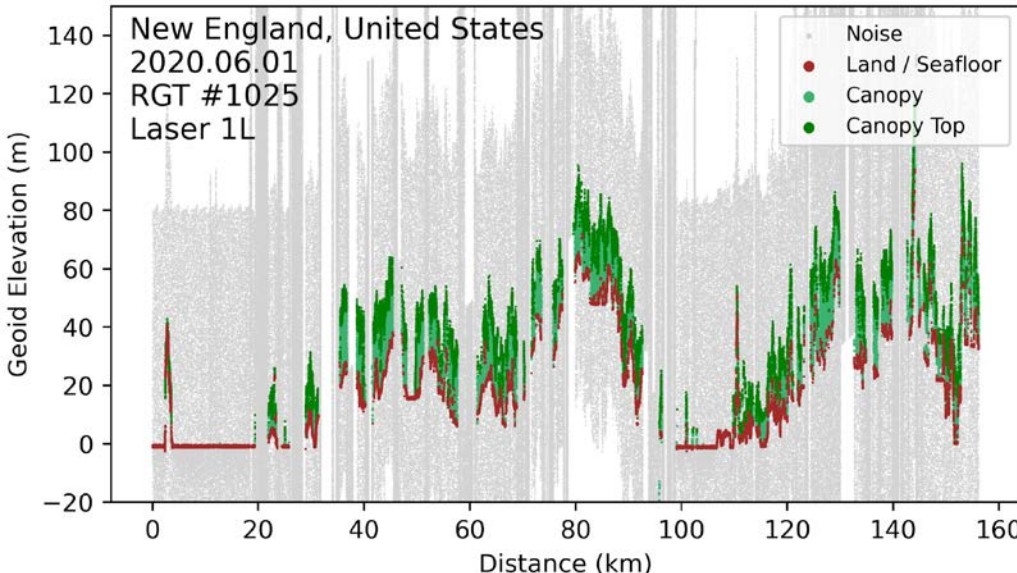

298

**Figure 3.** An ICESat-2 photon point cloud over New England, USA. Photons are classified to identify canopy, canopy-top, ground, and noise, according to filtering in the ICESat-2 ATL08 data product, and mapped at an individual photon level in ATL03 granules.

ICESat-2 granules are stored and archived at the NASA Distributed Active Archive Center and the National Snow and Ice Data Center (NSIDC). Data granules are formatted and distributed in orbit-track segments, where a single full earth orbit of the satellite is divided into 14 sub-segments by elevation band. While this format is useful when processing individual orbit paths (such as for producing Figure 3, above), it is inefficient for processing photons from multiple orbits that fall over an individual grid cell on a DEM. In those cases, large granule files must repeatedly be subset to extract the relatively small number of photons that lie within a specific grid cell, causing significant processing delays. The NSIDC DAAC provides a server-based subsetter for the data, but does not allow correctly combining the ATL03 and ATL08 datasets for photon classification, and thus was unusable for this project. To improve the performance of geospatial searches across multiple ICESat-2 granules, all ICESat-2 photons from calendar year 2021 were re-organized into geographic tiles. 417,660 tiles were created over the Earth's land surface at 0.25x0.25 degree boundaries, and photons from all granules collected in the calendar year 2021 were subdivided into data tiles for each target tile in which data was recovered.

ETOPO was validated on a cell-by-cell basis. First, each 15° ETOPO data tile was subset into 225 1x1° "sub-tiles" to reduce the total data load for each tile validation. For each 1x1° tile, a coastline validation mask was created using the CopernicusDEM dataset outlines, with water bodies and building footprints eliminated to ensure only bare-earth land elevations are being validated from ICESat-2. For each DEM cell, photons are collected falling within that grid cell. The top and bottom deciles (<10th and >90th percentile of z-elevations) of photons are eliminated to reduce the influence of outlier photons in the data.






With a spatial resolution of 15-arc-seconds (approximately 450 m at the equator), spatial sampling errors were seen to be
significantly skewing comparisons between ICESat-2 and DEM grid-cells. A grid-cell in a sloped or mountainous region, in
which ICESat-2 only "clips the corner" of a grid cell while missing a majority of the cell's spatial coverage (Figure 4, left),
can produce errors of tens to hundreds of meters between the grid-cell's "average" elevation and the average elevations of
ICESat-2 photons over the same grid-cell. To alleviate this spatial sampling bias, each 15-arc-second ETOPO grid well that
contained ICESat-2 data was subset in 15x15 1-arc-second subsets, photons were binned into each subset, and the total number
of subsets was tallied in order to compute a rough-order "coverage" estimate of ICESat-2 photons across an ETOPO grid-cell.
Figure 4 shows a schematic representation of this process, in which two grid cells with substantially different numbers of
ICESat-2 overlaps have differing coverage estimates.

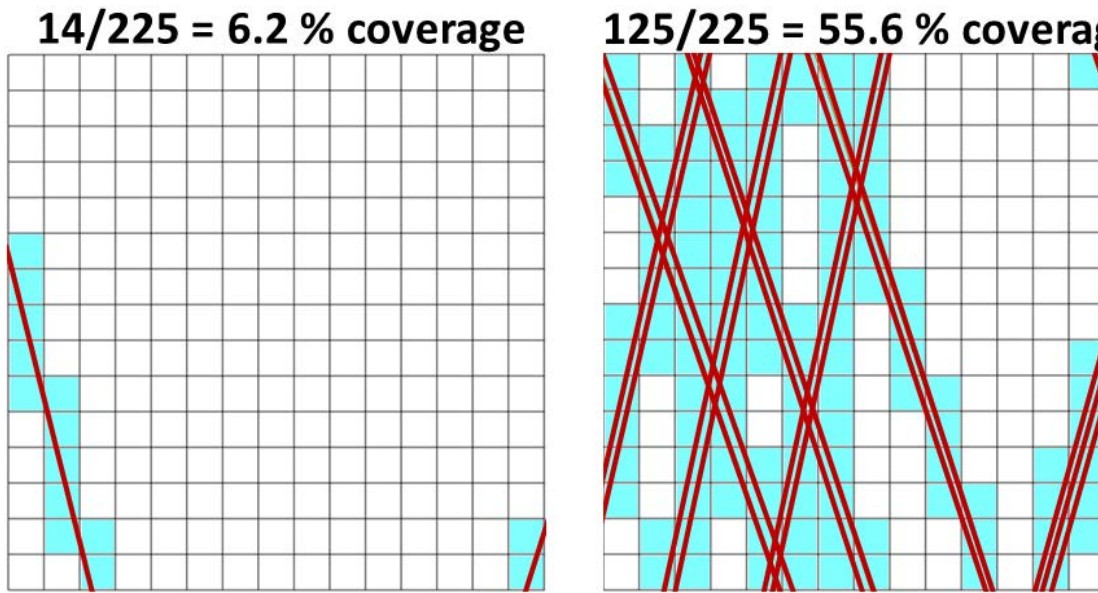


**Figure 4.** A schematic representation of two ETOPO grid-cells subdivided into 15×15 1-arc-second sub-cells to compute
cell coverage from ICESat-2 orbits. Left: A cell with only two partial orbit passes clipping the corners of the grid-cell,
with lower overall coverage. Right: A cell with multiple ICESat-2 orbit passes and higher coverage.

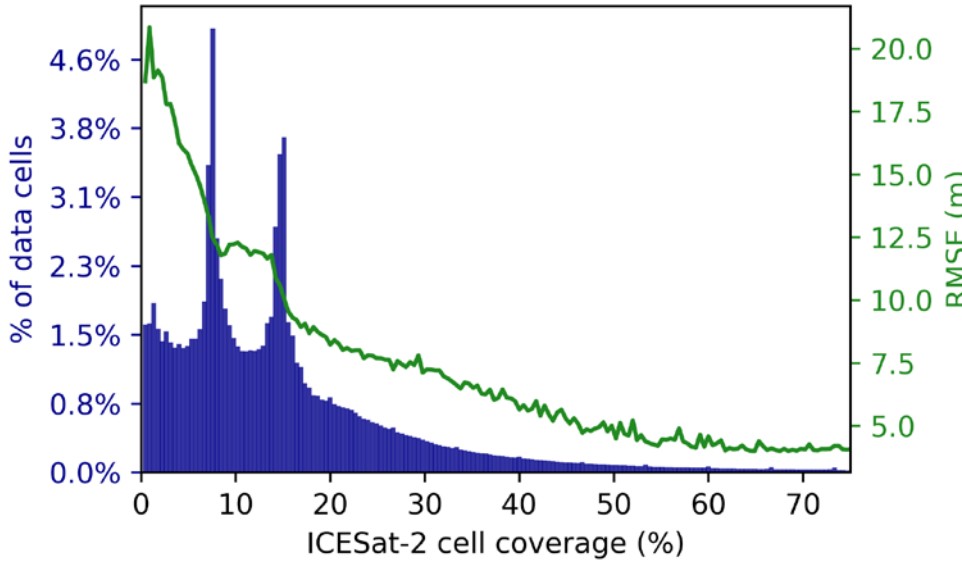

**Figure 5.** Distribution (blue bars, left) and RMSE (green line, right) of validated ETOPO grid cells as a function of ICESat-2 grid-cell coverage.

Errors were computed for each ICESat-2 grid cell by subtracting the ICESat-2-derived mean elevation of the grid cell against the ETOPO elevation. Figure 5 clearly shows the effect of spatial biasing, where grid cells that have significantly higher coverage estimates (~40% coverage) have consistently lower mean RMSE values compared to ICESat-2 estimates. In Figure 5, the two notable spikes in the histogram, at 7.5% and 15% coverage, correspond to ETOPO grid cells containing exactly one ICESat-2 orbit path, and exactly two orbit paths, respectively. Due to the converging orbits of ICESat-2 approaching its "pole hole" near 88 ° north and south latitude, a significant majority of ETOPO grid cells with higher ICESat-2 coverages (above 40%) are located in the polar regions, especially in Antarctica. This precluded using a set "minimum coverage" to filter out grid-cells with low coverage to calculate the RMSE of the ETOPO global dataset. Any such estimate would be dominated by validations predominantly over Antarctica. In order to avoid spatially biasing the validation data to the polar regions, while still eliminating lower-coverage grid cells that suffer from spatial sampling biases, we computed the RMSE of errors within each 1x1° sub-grid cell used for validation, and only chose grid-cells that had the top 5% coverage of all cells validated within that sub-tile. This provided validation data across a majority of Earth's land-surface (Figure 7, below) while minimizing errors introduced by spatial sampling biases, providing a "geographically weighted" estimate of ETOPO errors.

A small number of individual ICESat-2 granule files were found to have biased elevations relative to other orbits (even crossing orbits) in the same DEM tile, providing bimodal error distributions due to artifacts in one particular ICESat-2 granule. These specific ICESat-2 granules were flagged as anomalous data and omitted from further analyses.




Only the 288 ETOPO 15s tiles were validated in this manner. Since ICESat-2 cannot validate bedrock elevations underneath
the ice sheets, and only surface elevation tiles were validated. The ETOPO 30s and 60s global files were subsampled from
ETOPO 15s tiles, and were not independently validated.

**6 Validation Results**

Using the mean RMSE of the errors computed in grid-cells within each 1x1° ETOPO sub-tile, we find that ETOPO has a mean
RMSE over land of 7.24 m (Figure 6). Sub-tiles here are used in order to not geographically bias the validation data to the
poles, where more validation data exists. A map of these RMSE errors is provided in Figure 7. The geographic distribution of
errors clearly shows that RMSEs are greater in mountainous regions, a somewhat unsurprising result. The largest RMSE's
were seen at the coastline of Antarctica, where unavoidable mismatches can occur at the ice edge where consistently-calving
icebergs can open large leads and open water. ICESat-2 is measuring a constantly-changing surface while ETOPO is
attempting to represent a snapshot elevation dataset. Persistent negative biases of several meters are seen over the interior of
the Greenland and Antarctic ice sheets (where ETOPO showed lower elevations than indicated by ICESat-2) may be at least
partially an artifact of blowing snow caused by persistent katabatic winds, which is corrected for in ICESat-2's ATL06 Land
Ice Elevation (Smith and Team, 2023) data product, but was not used for these analyses because ATL06 version 5 did not
provide indices to map ice elevations back to a photon level as ATL08 does. ATL06 may be worked into future validation
efforts of other global DEMs beyond ETOPO 2022.

To our knowledge, this is one of the few instances where ICESat-2 has been used to validate a DEM on a global scale.

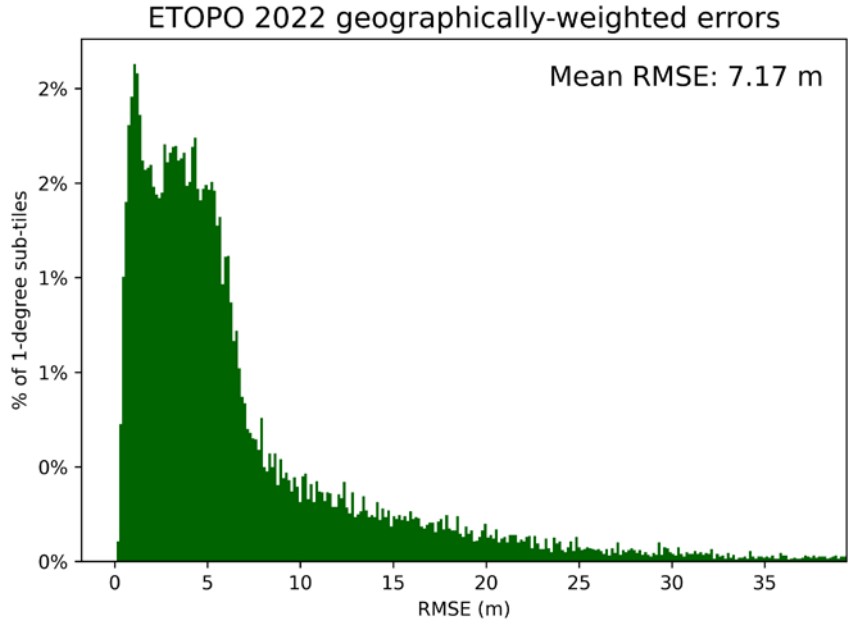


**Figure 6.** Distribution of ICESat-2 derived RMSEs averaged over each 1x1° ETOPO sub-tile over land. The mean RMSE of

the dataset is 7.17 m.

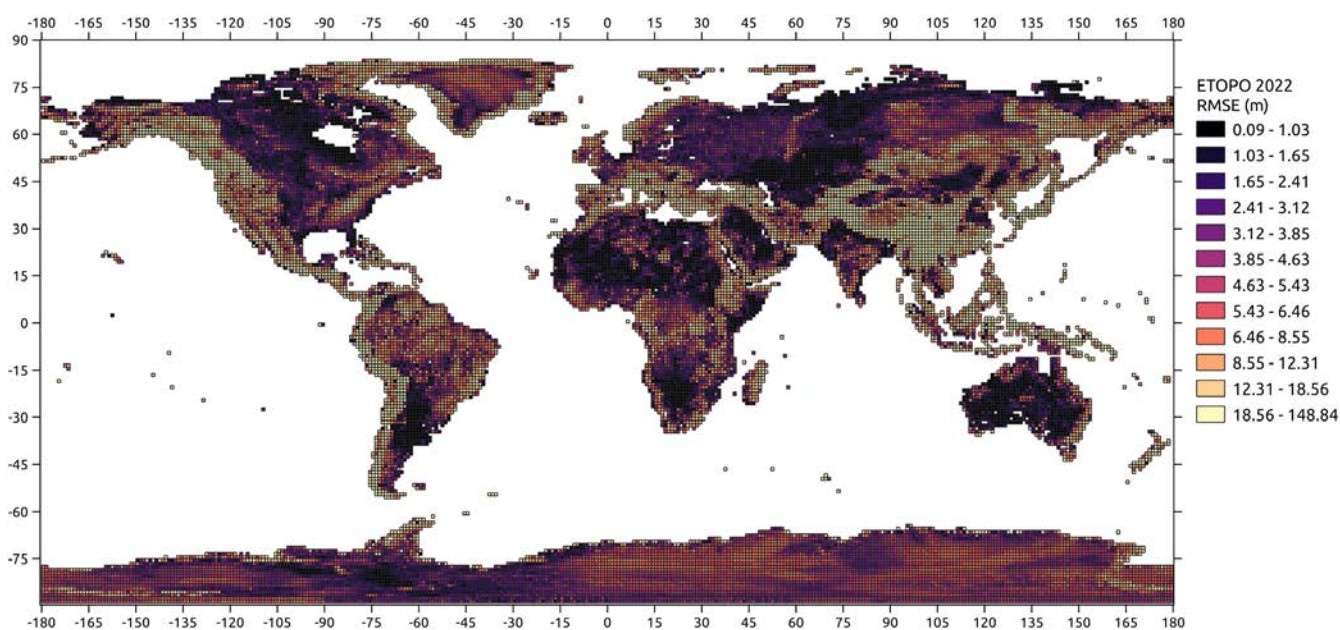


**Figure 7.** Map of RMSEs of 1x1° ETOPO sub-tiles over land, validated against ICESat-2.





## 7 Comparison with ETOPO1

ETOPO1, the previous iteration of NOAA's global seamless topographic-bathymetric Earth elevation data product, was released in 2010 at 1 arc-minute resolution, in both ice-surface and ice-bed versions (Amante and Eakins, 2009). Large amounts of elevation source data have been collected globally since ETOPO1's release, and as a result, ETOPO 2022 was built from entirely different datasets than ETOPO1, justifying a direct comparison. We compared the ETOPO 2022 60-second bed and surface grids to the ETOPO1 products on the same grid. Maps of the elevation differences are presented in Figures 8 and 9.

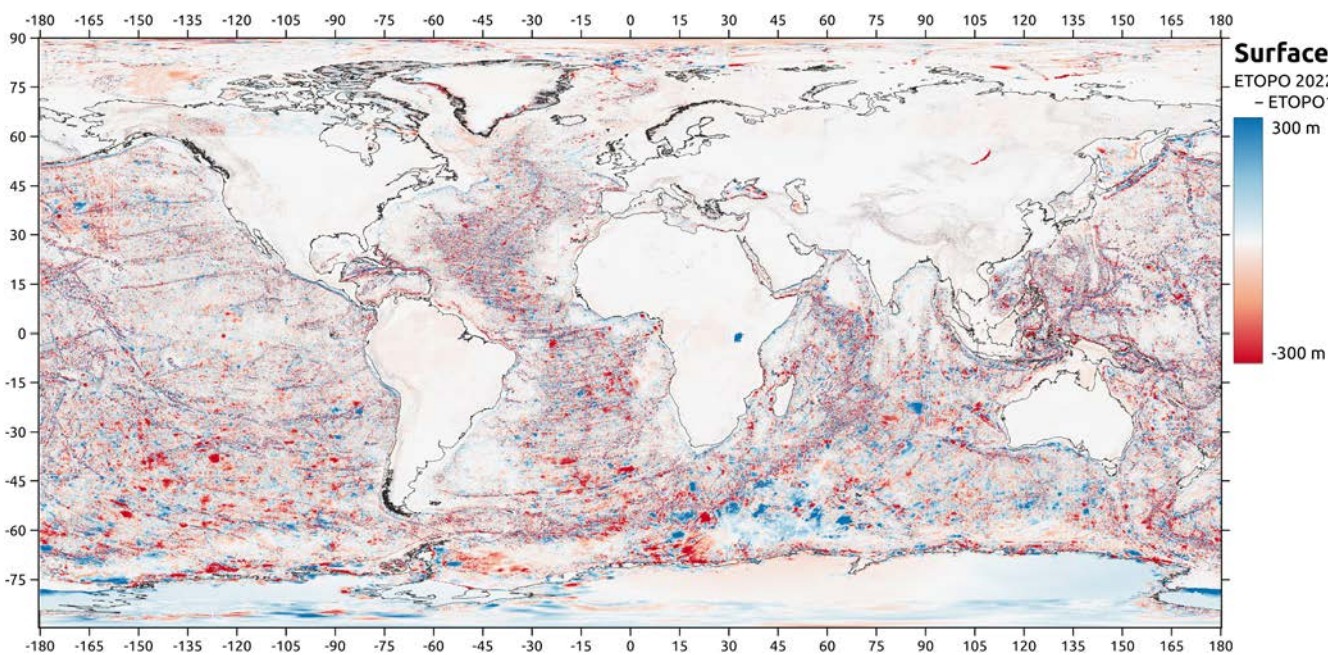

**Figure 8.** Map of elevation differences between ETOPO 2022 and ETOPO1, for ice surface datasets.

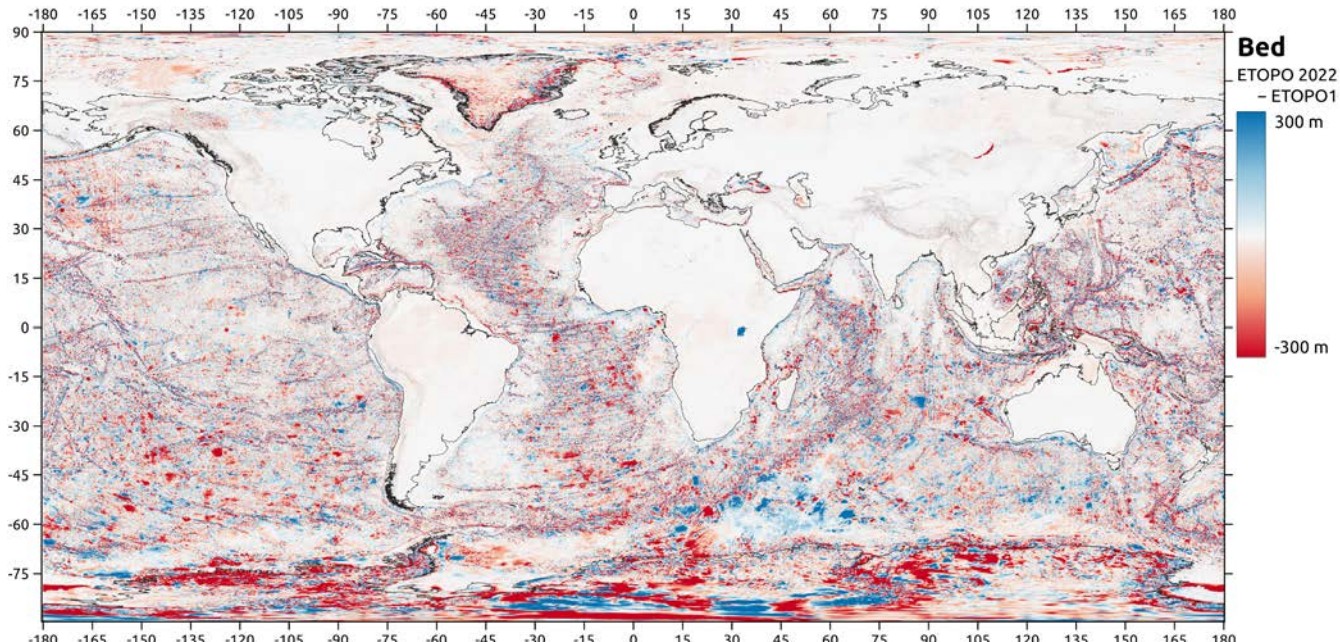

**Figure 9.** Map of elevation differences between ETOPO 2022 and ETOPO1, for ice bed datasets.

The greatest differences between ETOPO 2022 and the previous ETOPO1 product are in the ice sheet bed elevations (Figures 9 and 10.D), which had a root-mean-square (RMS) difference of 291 m from ETOPO1 to ETOPO 2022. The large discrepancies between these two datasets are a result of a vastly greater number of direct measurements of the ice sheet bed from ground-penetrating radar measurements, collected primarily via airborne measurements (MacGregor et al., 2021), and improved physically-based interpolations between depth measurements (Morlighem, 2020; Morlighem et al., 2017). Similarly, differences are large between the ocean bathymetries of the two datasets (RMS 152 m), owing to vastly greater volumes of bathymetry collected from new technologies such as swath-mapping multi-beam sonar. The differences are greatest in the Southern Ocean (Figure 9), where spaceborne gravimetric bathymetry estimates have improved our understanding of deep ocean bathymetry even where direct measurements remain sparse. Land elevation differences are relatively smaller (Figure 10.B, RMS 53.4 m). It is worth noting that in areas of heavy canopy cover, most notably in the Amazon and Congo rainforest basins, ETOPO 2022 records lower elevations than ETOPO1, largely due to the post-processing in FABDEM to reduce biases from canopy-top returns in spaceborne radar-altimetry collections. Also noteworthy is a visible "line" at 60 ° north latitude in northern Canada and Russia. North of this line the elevation differences between ETOPO1 and ETOPO 2022 are of markedly greater magnitudes (both positive and negative) than south of that line. Land surface elevations in ETOPO1 were primarily derived from NASA's Shuttle Radar Topography Mission (SRTM), first released in 2010, which only spanned up to 60° north latitude but excluded the polar regions. Elevations north of that line were derived by other methods, including lower-resolution spaceborne altimeters and digitized map data.



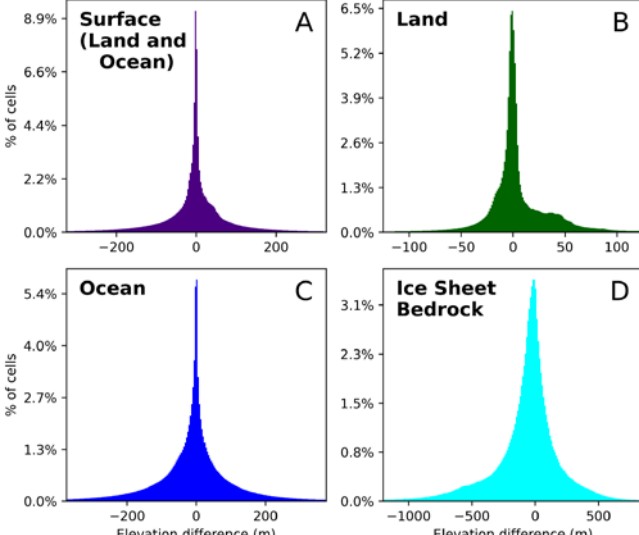

**Figure 10.** Histograms of ETOPO 2022 (60s) - ETOPO1 elevations, A) for all land and ocean surface elevations (Figure 9, full map), B) for land surface only. C) For ocean bathymetry only, and D) for ice sheet bed elevations (Figure 10, Greenland and Antarctic ice sheets). Note the different X-axes in the subplots.

Even with improved technologies, known issues exist in ETOPO 2022 that may be addressed in future versions. Large swaths of ETOPO 2022 ocean bathymetry come from the GEBCO data product, which itself comes from a wide variety of direct measurements and indirect interpolations. Since GEBCO and ETOPO 2022 use the same 15-arc second global grid, users who wish to see which source dataset GEBCO used in an ETOPO grid cell can download the GEBCO Type Identifier (TID) grids for accompanying GEBCO tiles (Mayer et al., 2018). Since many regions of the ocean floor remain unmapped by direct surveys, other methods are used to gap-fill direct measurements, such as inverse satellite gravimetry or interpolations between existing surveys. Especially close to the coast, such methods can produce artifacts such as deep "pits" of dozens-to-hundreds of meters depth in near-shore coastal regions, which may affect the accuracy of tsunami models and other use-cases in those regions (Amante and Eakins, 2016). We caution users when relying on GEBCO-derived near-shore bathymetry data to check the TID grids of the GEBCO surveys and pay attention to "indirect measurements" (TID #40-46) in those surveys.

ETOPO 2022 is not intended for navigational use, especially nautical navigation. Ships should rely upon coastal surveys and other bathymetric charts designed for navigational use.



## 8 Code and Data Availability

ETOPO tiles are freely available to use for all private, academic, or commercial purposes except navigation. Data is available for download on the NOAA ETOPO landing page: https://www.ncei.noaa.gov/products/etopo-global-relief-model. Source datasets for ETOPO are all publicly available at their respective data repositories outlined and referenced in Section 3. ETOPO data is covered by a Creative Commons Zero v1.0 Universal (CC0-1.0) license as described in NOAA's metadata description at https://data.noaa.gov/waf/NOAA/NESDIS/NGDC/MGG/DEM//iso/xml/etopo_2022.xml. When using ETOPO 2022 data from either link, please reference this manuscript as well as the following citation:

NOAA National Centers for Environmental Information. 2022: ETOPO 2022 15 Arc-Second Global Relief Model. NOAA National Centers for Environmental Information. https://doi.org/10.25921/fd45-gt74. Accessed [date].

A vast majority of processing for ETOPO 2022 was performed in Python 3.9, using open-source libraries and tools. Source code for the ETOPO workflow is maintained on its GitHub repository: https://github.com/ciresdem/ETOPO. The CUDEM suite of tools that ETOPO relies upon is maintained at its own repository: https://github.com/ciresdem/cudem. Both code repositories are covered by MIT open-access licenses (licenses viewable at each respective GitHub link).

The ETOPO 2022 User Guide is also available for download on the ETOPO landing page. Although this manuscript covers the processing steps in greater detail than the User Guide, the User Guide will be periodically updated whenever errors are found or revisions are made to the data, and is seen as the "most current" review of the dataset. The User Guide is a recommended reading for data users.

## 9 Competing Interests

The contact author has declared that none of the authors has any competing interests.

## 10 Acknowledgements

This research was supported by NOAA cooperative agreements NA17OAR4320101 and NA22OAR4320151. The Coastal DEM Team would like to specifically thank Kelly Stroker for her continued support and management to ensure the success of this project.



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
