# Peer review of "The Earth Topography 2022 (ETOPO 2022) Global DEM dataset"

_Earth System Science Data, 2024_

## Author Response (AR1)

**Review Comment 1: 'Comment on paper "The Earth Topography 2022 (ETOPO 2022) Global DEM dataset"', Anonymous Referee #1, 10 Sep 2024**

The authors provide a report on an open dataset of terrain and seafloor topography created at global level by integration of various existing products. The results are to be praised as a valid contribution towards integrating data, which can be used for future investigations, as these terrain/seafloor models are key for many applications. The authors correctly report on the methodology including the validation but not much is reported about the existing data merged together, its own accuracy metrics. It should be noted that the error distribution, propagation and thus the relative reliability of values in the final product are key information.

The work reports on integrating and validating a 15-arc-second topography and bathymetry dataset that consists in the collection of multiple other products. It provides "bare-earth" dataset, as reported in the abstract, thus a terrain digital model (DTM), as opposed to a digital surface model. We therefore expect the altimetric data to refer to the ground without vegetation and buildings. This should be noted as a limiting factor to some technologies is the canopy cover. NASA's ICESAT data are used to validate the altimetric values.

One thing to note is to assess reliability of the map, according to where the ICESAT values were used as validators, as the spatial distribution of high-quality elevation data might be biased to specific scenarios (e.g. low/high vegetation, flat terrain vs steep slopes etc…).  The challenge is two-fold: rigorous assessment of data also in complex scenarios, e.g. thick vegetation and/or mountainous terrains, and map reliability metrics to address areas were little information is available.

**Responses to Reviewer 1:**

***"The authors provide a report on an open dataset of terrain and seafloor topography created at global level by integration of various existing products. The results are to be praised as a valid contribution towards integrating data, which can be used for future investigations, as these terrain/seafloor models are key for many applications. The authors correctly report on the methodology including the validation but not much is reported about the existing data merged together, its own accuracy metrics. It should be noted that the error distribution, propagation and thus the relative reliability of values in the final product are key information."***

Thank you. We agree that a thorough investigation of the relative uncertainties of source datasets is ostensibly valuable. However, each of the 13 datasets ingested into ETOPO 2022 provides their own independent methods for validation and accuracy assessment,

using entirely different methods and reference datasets. Some of the source datasets (such as most bathymetric regions in GEBCO, most portions of BlueTOPO, and others) do not contain any validation or accuracy statistics at all. (This is not a criticism of those datasets. Much of the ocean floor remains entirely unmapped by direct measurements, performing independent validations where no independent measurements exist is an ongoing struggle for the bathymetric community.) Citing the stated statistics from each source dataset as-are would not provide an accurate comparison between them due to drastic differences in methods and data sources. It would result in an "apples to oranges" comparison providing more confusion than clarity. Also, this would inevitably prove misleading due to sampling biases. For instance, GEBCO provides a global coverage grid, but independent validation statistics of GEBCO are confined almost exclusively to land, much like ETOPO 2022. Meanwhile ETOPO 2022 primarily uses GEBCO data only over the oceans (Figure 1). As such, listing land-based validation statistics from GEBCO in a dataset table for ETOPO 2022 would provide almost no useful information relevant to its bathymetric use in ETOPO 2022, and would likely mislead the reader. A quantified intercomparison between all of the source datasets would be a valid experiment outside the scope of this paper, some of which has already been done in manuscripts independent to this work (e.g. Guth, P. L., & Geoffroy, T. M. [2021], https://doi.org/10.1111/tgis.12825). Due to these and other considerations, we instead focus our validation efforts on consistently assessing the outputs of the ETOPO 2022 product after resampling and combining the source data used. The accuracy metrics (and independent processing methodologies) of each of the 13 source datasets can be found in the references provided if readers wish to delve into those details.

> ***"The work reports on integrating and validating a 15-arc-second topography and bathymetry dataset that consists in the collection of multiple other products. It provides "bare-earth" dataset, as reported in the abstract, thus a terrain digital model (DTM), as opposed to a digital surface model. We therefore expect the altimetric data to refer to the ground without vegetation and buildings. This should be noted as a limiting factor to some technologies is the canopy cover. NASA's ICESAT data are used to validate the altimetric values."***

You are correct on all counts. Thank you for pointing this out so that we could make it more clear. We have added a sentence to Section "2.1 General Description and File Formats" explicitly stating the ETOPO 2022 dataset represents a DTM.

> ***"One thing to note is to assess reliability of the map, according to where the ICESAT values were used as validators, as the spatial distribution of high-quality elevation data might be biased to specific scenarios (e.g. low/high vegetation, flat terrain vs steep slopes etc…). The challenge is two-fold: rigorous assessment of data also in complex scenarios, e.g. thick vegetation***

*and/or mountainous terrains, and map reliability metrics to address areas were little information is available."*

ICESat-2 orbits the earth over 1387 reference ground tracks (RGTs), each of which repeat every 91 days. Due to ICESat-2's 92° orbital inclination, the RGTs converge near the poles and thus provide a positive sampling bias in the polar regions if no spatial resampling is performed. Our method explained in Sections 5 and 6 mitigates this latitude-based bias by binning results in 1x1-degree bins before averaging. (Otherwise ICEsat-2 data over Antarctica dominates the global validation results due to data volume alone.) Aside from latitude, there is no reason to believe that ICESat-2's 1387 RGTs would otherwise be geographically or spatially biased "*to specific scenarios (e.g. low/high vegetation, flat terrain vs steep slopes etc…)*".

However, we fully agree that ICESat-2 data presents multiple "two-fold challenge[s]" when processing data of one type (a vector-based profiling lidar) to validate data of another type (a gridded raster DEM) over challenging terrain. This data-type mismatch can affect the representative accuracy of validation data in "highly-sloped" terrains with greater elevation variability. Our approach of selecting only ETOPO grid cells with high levels of ICESat-2 "coverage" relative to nearby cells substantially mitigates the added uncertainty from spatial sampling bias but does not entirely eliminate it.

The performance of ICESat-2 validations over heavily-vegetated terrain is largely mitigated from NASA's ATL08 processing that filters out canopy vs. ground photons, along with ETOPO's relatively-coarse resolution allowing the collection of many photons within each cell. ICESat-2's 10-KHz laser pulses measure elevations every ~0.7 m along-track and can detect multiple photons per pulse, providing more than enough "ground" photons to accurately measure the mean ground elevation of an ETOPO grid cell even with dense canopy cover. For instance, over a 5x6-degree region of the Amazon rain forest with among the world's highest-density canopy cover (1-5 °S, 55-60 °W), after filtering for ground photons in only the "highest-coverage" grid cells, the mean number of ground photons used to calculate ground elevation was 559 photons per grid cell (minimum 207 photons), more than enough samples to statistically calculate a "mean altitude" within each cell from ICESat-2. The "high-coverage" thresholds minimized the spatial sampling bias within each grid cell by enforcing relatively widespread coverage across the grid cell's spatial footprint (rather than allowing ICESat-2 to just "clip the corner" of a grid cell to measure its elevation). Data volume over dense canopy can be a far greater issue if attempting to validate higher-resolution DEMs with cell footprints <0.1% the area of ETOPO 2022, where only a small handful of valid "ground" photons, if any, may be detectable within a single grid cell footprint. We took great care to navigate these challenges in order to produce as accurate of results as possible using ICESat-2 as a widespread and independently-measured sample of Earth's surface height. We fully agree with the reviewer however that this is an ongoing challenge. Our approach to it is described in the manuscript.

─────────────────────────────────────────────

**Review Comment 2: 'Comment on essd-2024-250', Anonymous Referee #2, 07 Oct 2024  reply**

I commend the authors for producing this data set, and the comparison with ICESat-2.  They convincingly demonstrate that the new data sources (and probably improved techniques) greatly improve on the older version of ETOPO.  This data set directly matches SRTM15+, which should be acknowledged.  There should be a direct comparison, and short discussion of the differences, between ETOPO and SRTM15+.  I would suggest a map like Figure 9 for the differences, and an RMSE map like Figure 7.

I commend the data set (or its creators) for doing the following, which is unfortunately not universally done with similar data sets.

Including the version number of the DEM in the file names

Including the vertical datum with Geotiff key 4096 inside the file's metadata.

Lines 165-170: Is this data set publicly available?  I checked the PE&RS paper, and can't find any reference on where to actually get the data.  Most of the data sets I am familiar with, but it would be nice to have explicit download links for all of these which are public.

Line 180: how was the reprojection done?  Going from the US projections to EGM is non-trivial.  This is covered later in section 4.2.  I think VDATUM only works in the US, so that limitation might be mentioned.

Line 185: please provide the link.

Line 332, Figure 4: is the difference between the two cells due to latitude (higher coverage at high latitude), clouds, or something else?  The coverage would of course be greatly improved by using additional years data.

Line 359: delete "and" (the 4th word)

Line 380, figure 7: the colors on the figure are very hard to read.  Can the color palette be improved?  Or use two figures with different ranges, one for the low RMSE, and another to show the large ones?

Line 439: put in date

Comments for the authors to think about for future work, which do not necessarily need to be addressed for this paper:

There are now many more years of ICESat-2 data, so Figures 4 and 5 could be greatly improved, but that is not a reason to revise the manuscript.

The following comments are based on my work with DEMs, most importantly the 1" global data sets. I include them here so the ETOPO team can think about them, as the 1" DEMs are generalized and the ETOPO moves to smaller and smaller grid spacings, I see issues arising:

I would prefer that the tile names start at the SW corner, which I think is more common with global DEMs, or include the full range as is done with the AW3D30 (N000W060_N005W055, admittedly for zipped directories, but could be done for tiles). One can of course keep track of the peculiarities of each DEM, but it would be nice for all of DEMs to be consistent, and for a quick glance at a file to be unambiguous).

Most of the 1" DEMs have the pixel centroids aligned on parallels and meridians; a few have a ½ pixel offset. This is more than just the pixel-is-area or pixel-is-point, but is also affected by the starting grid corner. This means that point elevations cannot be directly compared (for instance between CopDEM and AW3D30) without reinterpolation of one. It would be nice to agree on the pixel representation moving forward, and have a consistent standard.

**Responses to Reviewer 2:**

*I commend the authors for producing this data set, and the comparison with ICESat-2. They convincingly demonstrate that the new data sources (and probably improved techniques) greatly improve on the older version of ETOPO.*

Thank you sincerely for the commendation.

*This data set directly matches SRTM15+, which should be acknowledged. There should be a direct comparison, and short discussion of the differences, between ETOPO and SRTM15+. I would suggest a map like Figure 9 for the differences, and an RMSE map like Figure 7.*

We've added a short paragraph to the end of Section 2.1 that indicates the data is sampled within a geometrically equivalent grid to both SRTM15+ and GEBCO, although the source datasets and processing techniques are different. However, displaying a global differencing map between ETOPO 2022 and SRTM15+ and GEBCO is somewhat outside the scope of this paper and we are not sure it would be particularly illustrative with just a couple of figures. More detailed analyses would be needed to make it informative to readers. Over land, both SRTM15+ and GEBCO (which uses NASADEM over land, based on SRTM) are primarily sourced from NASA's Shuttle Radar Topography Mission collected in February 2000. ETOPO1 also used SRTM as its primary land-cover dataset. We opted to present a brief comparison with ETOPO1 simply because the ETOPO 2022 dataset is presented here as a direct upgrade to the older ETOPO1 data. A rigorous intercomparison between all four

datasets (ETOPO 2022, ETOPO1, SRTM15+, and GEBCO), including independent validations of each against ICESat-2, would quickly balloon into a very labor-intensive endeavor, likely expanding the paper's length by more than the rest of the paper combined. (Notably, the processing of ICESat-2 data for ETOPO 2022 validation took nearly 3 months processing on a 20-core machine. Re-doing the validation for all global datasets would delay the publication of this paper substantially.) It also opens broader questions of "how best to validly assess" each dataset given the 15-20 years spanning data collection times from each respective source. We feel that's somewhat outside the scope of this paper, whose primary purpose is to present the ETOPO 2022 product as it was built. Our added paragraph indicates this is a good exercise for a future paper and we acknowledge that opportunity.

> *I commend the data set (or its creators) for doing the following, which is unfortunately not universally done with similar data sets.*
> *Including the version number of the DEM in the file names*
> *Including the vertical datum with Geotiff key 4096 inside the file's metadata.*

Thank you. Like you, we have worked with many datasets for which these parameters are ill-defined. We attempted to ensure useful metadata is readily accessible.

> *Lines 165-170: Is this data set publicly available?  I checked the PE&RS paper, and can't find any reference on where to actually get the data.  Most of the data sets I am familiar with, but it would be nice to have explicit download links for all of these which are public.*

Thank you, agreed. Yes it is publicly available. We added a link to the Shallow Bathy Everywhere website where it can be accessed online, as well as made a couple of specific clarifications in the text there.

> *Line 180: how was the reprojection done?  Going from the US projections to EGM is non-trivial.  This is covered later in section 4.2.  I think VDATUM only works in the US, so that limitation might be mentioned.*

Yes, thank you for pointing out this section did not make explicitly clear how these conversions were performed. We believe our description in section 4.2 covers the methodology used for these conversions, and is a better place to outline those methods than in the "BlueTopo" section (3.6). We agree and acknowledge these conversions are a non-trivial endeavor. In section 4.2 we've added an additional reference to the CUDEM manuscript (Amante, et al, 2023) that describes the CUDEM team's "vdatums" module that was used when processing source datasets for ETOPO, and we clarified some of the writing in that section. "vdatums" incorporates processing from the similarly-named NOAA/NOS "VDatum" module ([https://vdatum.noaa.gov/](https://vdatum.noaa.gov/)), which itself incorporates NOAA's

"htdp" (horizontal time-dependent positioning) and the National Geodetic Survey's (NGS) "NCAT" (NGS Coordinate Conversion and Transformation) tools. "vdatums" also includes processing of EGM and other global vertical grids, as well as localized tidal datums. More information about the CUDEM "vdatums" module can be found in the Amante, et al, (2023) reference in the paper, and on the team's GitHub repository at https://github.com/ciresdem/cudem/blob/main/docs/vdatums.md.

Of note: the NOAA/NOS VDatum tool is not limited solely for use over US lands and coastal waters; VDatum includes multiple globally-defined horizontal and vertical datums. The "navd88" survey datum specifically mentioned in the BlueTopo section is indeed confined to North America and coastal waters, and the NOAA BlueTopo dataset is constrained within the bounds of the navd88 reference frame.

> *Line 185: please provide the link.*

Thank you for the recommendation. We've included a link to the BOEM "Deepwater Gulf" dataset. The editors can decide whether the link is more appropriate in the main text or in the References section.

> *Line 332, Figure 4: is the difference between the two cells due to latitude (higher coverage at high latitude), clouds, or something else? The coverage would of course be greatly improved by using additional years data.*

Figure 4 is a representative illustration, but the answer to your question is "all of the above." Grid cells at higher latitudes (where the ICESat-2 orbits converge) on average have far greater coverage than grid cells closer to the equator. Cloud cover unavoidably limits data collection over some grid cells more than others. Additionally, orbital geometry dictates, predictably, some grid cells will simply have more ascending and/or descending satellite overpasses than others. ICESat-2's 1387 reference ground tracks (RGTs) are spaced approximately 12.5 km apart at the equator, with 3 km between each of the three strong lasers, and 90 m between each strong-weak laser pair. Not all ETOPO grid cells receive any ICESat-2 coverage at all, while others receive very little. Figure 4 is meant to illustrate how coverage was quantified within each individual ETOPO grid cell in order to reduce sampling biases by eliminating grid cells that have sparse validation coverage, based on the trend seen from the data in Figure 5. Due to the repeating RGTs, more years of ICESat-2 data would increase computational costs but not necessarily "greatly improve" overall coverage. (We give a little more detail in response to your comment below.)

> *Line 359: delete "and" (the 4th word)*

Thank you! Done.

*Line 380, figure 7: the colors on the figure are very hard to read. Can the color palette be improved? Or use two figures with different ranges, one for the low RMSE, and another to show the large ones?*

We chose the "Viridis Inferno" color palette for this figure because of its perceptually-uniform color gradient as well as its compatibility for colorblind readers. Additionally we chose an equal-volume scaling for the color-bins rather than a linear (equal-interval) scaling, to make the distribution of cells more visually legible given the non-uniform and non-gaussian distribution of the underlying data (Figure 6). We feel this was an appropriate choice and effectively conveys the distribution of errors as presented on a global scale. While other finer-grained colormaps would be preferable when zooming in to individual regions, including flatter "low-RMSE" regions, we feel that separating the same global map into two different maps with different color scales would likely confuse the presentation of the data rather than clarify it for a majority of readers.

*Line 439: put in date*

Line 439 provides an example of how to cite the ETOPO 2022 dataset. The "[date]" field is to be filled in by the data user and is intentionally omitted here.

*Comments for the authors to think about for future work, which do not necessarily need to be addressed for this paper:*

We appreciate the reviewer's insightful comments below. Although we may not be required to respond for the scope of this review, we would like to offer responses to these comments in the spirit of open community discussion.

*There are now many more years of ICESat-2 data, so Figures 4 and 5 could be greatly improved, but that is not a reason to revise the manuscript.*

We used a full year of ICESat-2 data (Jan-Dec 2021) covering the calendar year immediately prior to publication of ETOPO 2022, which provided nearly a trillion post-filtered photons for validation. More years of ICESat-2 data would indeed provide more photons but would not inherently change the nature of the validation nor significantly improve figures 4 or 5. Figure 4 is a conceptual representative drawing. Figure 5 shows a well-defined distribution of grid-cell coverages and trends in accuracy based on those coverages. The shape of the data distribution in Figure 5 would likely only marginally change with more years of ICESat-2 data. ICESat-2 performs repeat passes covering 1387 Reference Ground Tracks (RGTs) every 91 days. Roughly speaking, ICESat-2 "covers the globe" 4x per year. More years of data would slightly improve coverage over tracks that may have been cloud-covered before, but would not significantly improve the validation results, and may in fact create issues due to temporal changes in physical topography spanning multiple years.

Adding more years of ICESat-2 would not solve the underlying geometric issue that some ETOPO grid cells will inevitably have "high coverage" of intersecting ICESat-2 orbit paths while others nearby have little-to-none. Although we agree that—generally speaking-—"more data is better", in this case it quickly reaches a point of rapidly-diminishing returns while linearly increasing computational costs.

> *The following comments are based on my work with DEMs, most importantly the 1" global data sets.  I include them here so the ETOPO team can think about them, as the 1" DEMs are generalized and the ETOPO moves to smaller and smaller grid spacings, I see issues arising:*
> *I would prefer that the tile names start at the SW corner, which I think is more common with global DEMs, or include the full range as is done with the AW3D30 (N000W060_N005W055, admittedly for zipped directories, but could be done for tiles). One can of course keep track of the peculiarities of each DEM, but it would be nice for all of DEMs to be consistent, and for a quick glance at a file to be unambiguous).*

We agree there is always a tension about the best way to reference the "origin" of DEM grids. Longstanding computational norms put the row-column "origin pixel" of raster files (including GeoTiffs and NetCDFs) in the upper-left corner of the screen, which makes the directional naming of ETOPO's tiles consistent with the spacing of the grid-cells within each tile and its embedded "geotransform" data structures. Thus, the "outer" (tile-naming) and "inner" (coordinate) grids follow the same consistent axes. This is conceptually and computationally consistent for those working with the data within these files. We certainly appreciate your point that a "bottom-left" naming convention as what AW3D30 chooses to do is more consistent with a lat-lon coordinate grid and may be more intuitive at first glance. Conversely, this puts the row-col coordinate grids within each file in a different orientation than the naming convention of the files themselves. This tension has been present in geographically-based rasters since the dawn of digital remotely-sensed images and is unlikely to go away with any release of ETOPO. Like you, we wish we had a universal solution.

> *Most of the 1" DEMs have the pixel centroids aligned on parallels and meridians; a few have a ½ pixel offset. This is more than just the pixel-is-area or pixel-is-point, but is also affected by the starting grid corner.  This means that point elevations cannot be directly compared (for instance between CopDEM and AW3D30) without reinterpolation of one.  It would be nice to agree on the pixel representation moving forward, and have a consistent standard.*

Similar to above, this is a longstanding tension within the DEM and remote-sensing community and is unlikely to be resolved with any release of ETOPO. The CopernicusDEM,

for instance, uses grid-center referencing along coordinate grids while GEBCO uses grid-corner referencing. CopernicusDEM uses "pixel-is-point" representation while GEBCO uses "pixel-is-area." We acknowledge, some add a "½-pixel offset" onto the geotransform objects in the file headers that (at times) complicate data processing depending upon the tools being used. Each standard has valid arguments for its "superiority" compared to the other.

We processed ETOPO 2022 to calculate the "mean" elevation of data contained within a grid-cell ("pixel-is-area"), which lends itself to grid-corner referencing that defines the boundary rather than the center of a given cell. A grid cell with its center on the north or south pole would otherwise have an ill-defined cell boundary extending a half-pixel 'above' or 'below' each respective pole. A "½-pixel offset" could fix this but then adds a complication that many software packages do not correctly handle these ½-pixel offsets contained in dataset headers. (Conversely, there are drawbacks to the "pixel-is-area" and "grid-corner" approach used in ETOPO that we could expound upon in detail, but our response here is already growing lengthy.)

We dealt with these inconsistent standards repeatedly among source datasets while processing input datasets for ETOPO 2022. We agree that having a consistent standard everyone agrees to use would immensely simplify the work of the global community, but is not likely possible given the wide variety of processing methods and use cases. In the spirit of professional humor we offer the popular XKCD comic "Standards": https://xkcd.com/927/
* * *
**Additional notes to editor:**
We sincerely thank both anonymous reviewers for their inputs and suggestions, many of which strengthened the manuscript and/or made us question the assumptions within our work. And to the editors for their extended patience through the entire process. We offer our sincere gratitude.

One note we (the author team) would like to make about a particular typo that the reviewers understandably missed. It was toward the end of the paper and was easily overlooked. We did not see it in fact until just recently, upon re-reading the manuscript to reply to anonymous reviewers.

The opening sentence of Section "6 Validation Results" in the pre-print contains a typo, presenting the mean RMSE over land as 7.24 m, which differs slightly from the 7.17 m result listed in both the Abstract and in Figure 6. This was a mistake we missed from earlier edits of the initial manuscript, which was subsequently not caught by either reviewer. We checked our work against the data and have corrected the typo in Section 6 for consistency and correctness. The correct value is 7.17 m as stated in the abstract.

---

## Author Response (AR2)

**Response to Editor:**

Thank you, we have received Reviewer #3's comments, which are helpful and insightful, and made minor revisions accordingly that we think improve the manuscript. We also found a small handful of additional typos and corrected them (all changes are tracked in the appropriate submitted document), as well as a few errors in our references list that crept in while editing. Please let us know if there is anything else needed, and thank you so much.

**Response to Reviewer 3:**

Hello,

We thank our third reviewer both for their positive remarks, and for their genuinely insightful comments and suggestions. These were extremely helpful and we've taken them to heart.

**line 43: \*is\* intended**

Fixed, thank you! It's always amazing how even after dozens of sets of eyes, typos can still be found.

**57: rather "... is not the focus of the current work." or similar. "furure work" should rather appear in a perspectives section/paragraph towards the end of the manuscript.**

Thank you. We elected to keep this paper to a dataset description only and omit a "future work" section. This sentence was added in response to a previous reviewer's suggestion, but upon reread it is likely not necessary to even mention. We have omitted that sentence and will keep the focus on the current dataset.

199: Copernicus DEM being from altimetric radar. While conceptually correct, this is potentially misleading. "Radar altimeters" are typically nadir looking. The Copernicus DEM is mainly (not only) from the TanDEM-X radar interferometry mission, a different technique than altimetry (as typically understood in the remote sensing community).

You're completely right on all counts, and thank you so much for pointing that out. Our statement about Copernicus data was oversimplified as stated there, and unintentionally misleading. We deleted the phrase "from spaceborne altimetric radar measurements" (which you're right, although not stating it, can by-default imply nadir Jason-style radar altimeters, of which TanDEM-X is not). We replaced it with "primarily using the TanDEM-X synthetic aperture radar." We hope this conveys the basics more clearly and accurately while not drowning the reader in too much detail about Copernicus processing (which they can go to the CopernicusDEM, 2022 reference if they wish to read more detail).

381 and following: the difference between ICESat-2 and Copernicus DEM over ice can have many other reasons than blowing wind. E.g. elevation change over time (the Cop DEM consists mainly of TanDEM-X in the early 2010s; TanDEM-X is an X-band radar mission with potential penetration into the firn of several metres.

Thank you, you're right, we mentioned one potential source of these biases but omitted others, which in retrospect is potentially a greater cause of the biases. We have added a sentence to address that before one that mentions blowing snow.

Although glacial drawdown may play a role as well over outlet glaciers, these biases were widespread enough (across the interior of both the Greenland and Antarctic plateaus) that they are probably not the cause of this.

**Which direction has the bias you mention?**

It's a negative DEM bias, indicating ETOPO surface was several meters beneath what ICESat-2 was saying in those specific regions. This direction is consistent both with snow-penetration effects (which will lower the stated DEM surface below the "true" surface) and blowing snow (which would raise the elevation of ICESat-2 returns above the "true" surface). We did not have the necessary data to do a full sub-study into the relative effects of each, but we mention them both in the results here to make the reader more aware. Thank you, this was genuinely helpful.

**Fig. 7 is interesting. At least in the jpg-compression the grid lines become quite thick and disturb the RMSEs. Make the grid lines thinner, or even remove them?**

Excellent Point. We replaced Figure 7 with an identical figure that has the grid lines much thinner and light-grey (almost invisible), which does make it noticeably easier to read. We want to keep some hint of the lines to avoid mid-ocean "island" cells from being nearly invisible in the image.

Optional, the authors might want to consider a short (sub-)section about "known limitations" for the use of the data set. This might be quite useful and important for users, as some might use the data set in unanticipated ways.

Thank you, we have added a section "8 Known Issues and Limitations" that I think will help the reader in the ways you described, and put a couple paragraphs that may help guide and warn users about potential pitfalls and suggestions for each.

Again, thank you for your time and comments, they were genuinely helpful.